# Association between the Triglyceride-Glucose Index and Vitamin D Status in Type 2 Diabetes Mellitus

**DOI:** 10.3390/nu15030639

**Published:** 2023-01-26

**Authors:** Qunyan Xiang, Hui Xu, Junkun Zhan, Shuzhen Lu, Shuang Li, Yanjiao Wang, Yi Wang, Jieyu He, Yuqing Ni, Linsen Li, Yiyang Liu, Youshuo Liu

**Affiliations:** 1Department of Geriatrics, The Second Xiangya Hospital of Central South University, Changsha 410011, China; 2Institute of Aging and Age-Related Disease Research, Central South University, Changsha 410011, China; 3Department of Nursing, Hunan Normal University School of Medicine, Changsha 410013, China

**Keywords:** triglyceride-glucose index, vitamin D deficiency, type 2 diabetes mellitus, insulin resistance

## Abstract

Background: Vitamin D deficiency (VDD) increases the risk for type 2 diabetes mellitus (T2DM), which might be related to insulin resistance (IR). We aimed to explore the association between the triglyceride-glucose (TyG) index, a reliable indicator of IR, and VDD in patients with T2DM. Methods: There were 1034 participants with T2DM enrolled in the Second Xiangya Hospital of Central South University. The TyG index was calculated as ln (fasting triglyceride (TG, mg/dL) × fasting blood glucose (mg/dL)/2). VDD was defined as 25-hydroxyvitamin D [25(OH)D] level <50 nmol/L. Results: Correlation analysis showed a negative association between the TyG index and 25(OH)D level. After adjustments for clinical and laboratory parameters, it was revealed that when taking the Q1 quartile of TyG index as a reference, an increasing trend of VDD prevalence was presented in the other three groups divided by TyG index quartiles, where the OR (95% CI) was 1.708 (1.132–2.576) for Q2, 2.041 (1.315–3.169) for Q3, and 2.543 (1.520–4.253) for Q4 (all *p* < 0.05). Conclusions: Patients with higher TyG index were more likely to have an increased risk of VDD in T2DM population, which may be related to IR.

## 1. Introduction

Diabetes mellitus is a severe, chronic disease posing an enormous threat to public health and an economic burden worldwide. Globally, diabetes mellitus prevalence was estimated at 10.5% (536.6 million people) in 2021, and this number will rise to 12.2% (783.2 million people) in 2045 [1]. Type 2 diabetes mellitus (T2DM) is a low-grade inflammatory and age-related disease, accounting for more than 90% of diabetes mellitus. The pathogenesis of T2DM is complex and believed to be closely related to insulin resistance (IR) and subsequent dysfunction of pancreatic beta cells. Recently, emerging evidence has demonstrated that Vitamin D may contribute to T2DM pathogenesis by increasing insulin sensitivity and maintaining pancreatic beta cell function [2,3,4,5].

Vitamin D is a pleiotropic hormone, which is primarily composed of vitamin D2 (ergocalciferol) and vitamin D3 (cholecalciferol). Vitamin D from the skin or diet needs to be first hydroxylated into 25-hydroxyvitamin D [25(OH)D] and then be hydroxylated into 1, 25-dihydroxyvitamin D [1,25(OH)2D] [6]. The main biological functions of vitamin D in regulating calcium and phosphorous homeostasis has been fully elucidated [7]. Besides, vitamin D receptor (VDR) has been detected in a variety of tissues, which has led to an increasing recognition of its function in cell proliferation, terminal differentiation, angiogenesis, inflammatory and immune response [8,9]. Recently, it has been identified that VDR is presented in pancreatic beta cells, adipose tissue and skeletal muscle that have a close relationship to insulin responsiveness, and thus contribute to the pathogenesis of T2DM [4,10,11,12]. Moreover, some studies from observational and longitudinal cohorts reported that serum 25(OH)D level was inversely related to IR and T2DM and increased the risk of T2DM incidence [13,14,15,16,17].

The triglyceride-glucose (TyG) index, calculated as ln (fasting triglyceride (TG, mg/dL) × fasting blood glucose (mg/dL)/2), is a new and credible indicator of IR [18,19]. A growing body of researches demonstrated that the TyG index was associated with newly diagnosed T2DM [20,21,22,23]. Besides, it was also related to cardiovascular outcomes [24] and diabetic complications [25] in diabetic patients. In a previous study, serum vitamin D levels of male patients with T2DM were negatively associated with TyG index but not those of female patients [26]. However, their study included a relatively small number of participants, and the conclusions need to be further verified. The present study was conducted to explore whether there was an association between the TyG index and vitamin D deficiency (VDD) in patients with T2DM. In addition, we intended to evaluate if the TyG index could be used as a predictive tool for VDD prevalence.

## 2. Materials and Methods

### 2.1. Study Population

We enrolled a total of 1034 participants with T2DM, including 621 men and 413 women, in the Second Xiangya Hospital of Central South University from September 2020 to October 2022. T2DM was diagnosed based on American Diabetes Association criteria [27]. The exclusion criteria were: (1) type 1 diabetes mellitus or other types of diabetes mellitus; (2) pregnancy or lactation; (3) acute diabetic complications, acute inflammatory diseases, malignant tumor, stroke or myocardial infarction history; (4) parathyroid diseases, nephrolithiasis, autoimmune nephrosis or other severe renal diseases with the estimated glomerular filtration rate (eGFR) <15 mL/min/1.73 m^2^; (5) Use of medications affecting the synthesis or metabolism of vitamin D, such as phenytoin, phenobarbitone, isoniazid, and glucocorticoid. The Ethics Committee of the Second Xiangya Hospital of Central South University approved this study ((2022)085). Besides, the study was carried out in accordance with the Declaration of Helsinki and received written informed consent from each participant.

A flowchart presenting the procedure for assigning patients to the appropriate groups is shown in Figure 1. Hence, 132 patients missing information of key variables (serum levels of 25(OH)D, fasting triglyceride, glucose, etc.) were excluded. In addition, 151 patients with acute diabetic complications, some severe diseases, and other diseases affecting the analysis were excluded. Hence, 1034 participants were included for the present analysis.

### 2.2. Data Collection and Definition

The baseline clinical characteristics (including sex, age, smoking, a history of hypertension, usage of medications, such as antidiabetic drugs and statins) were collected by a detailed self-report questionnaire at admission. During the physical examination, experienced caregivers measured the patient’s body height, weight, and blood pressure. Body mass index (BMI) was calculated as weight in kilograms divided by height in meters squared (kg/m^2^) was used to calculate. According to the recommendation from the Group on Obesity in China, obesity was defined as BMI ≥28 kg/m^2^ [28]. Hypertension was defined as repeated measurements of systolic blood pressure (SBP) and/or diastolic blood pressure (DBP) ≥140/90 mmHg, a history of hypertension, or taking antihypertensive medications in the previous two weeks [29]. Cigarette smoking was considered as smoking cigarettes continuously or cumulatively for more than 6 months [30]. VDD was defined as the serum 25(OH)D level <50 nmol/L [31,32,33,34]. Otherwise, a serum 25(OH)D level ≥50 nmol/L was classified as no VDD controls.

### 2.3. Biochemical Parameter Detection

Blood samples were obtained from each participant after fasting for 8–12 h using standardized sterile tubes, and were then centrifugated at 2500× *g* for 15 min. Biochemical parameters, including triglyceride (TG), total cholesterol (TC), low-density lipoprotein cholesterol (LDL-C), high-density lipoprotein cholesterol (HDL-C), and fasting glucose were determined using a HITACHI 7170A analyzer (Instrument Hitachi Ltd., Tokyo, Japan) [35]. Other serum parameters, including C-peptide, Albumin, Uric acid (UA), blood urea nitrogen (BUN), and serum creatinine (Scr), were measured using an ARCHITECT c8000 System (Abbott Laboratories, Irving, TX, USA) [36]. Serum levels of glycated hemoglobin (HbA1c) were measured using a Hemoglobin Testing System (VARIANT-11, Bio-Rad, Hercules, CA, USA). The serum concentration of 25(OH)D was measured using a chemiluminescence assay (Siemens ADVIA Centaur XP, Burlo, Germany). Urine samples were obtained in the morning and then centrifugated at 2000× *g* for 10 min. Immunoturbidimetry (Beckman Coulter, Brea, CA, USA) was used to measure albumin and creatinine in the urine before calculating the urinary albumin-to-creatinine ratio (UACR). Calculation of the eGFR was performed using the equation from the Modification of Diet in Renal Disease Study [37]. The TyG index was calculated as ln (fasting triglyceride (mg/dL) × fasting glucose (mg/dL)/2).

### 2.4. Statistical Analysis

The quantitative data that were normally distributed were described as the mean ± standard deviation (SD), whereas the parameters of abnormal distributed were presented as median with interquartile ranges (25–75%). The qualitative parameters were presented as the numbers and percentages. Comparisons of the quantitative variables between the VDD group and no VDD control (CON) group were analyzed by Student’s *t*-test or non-parametric Mann–Whitney U test according to the distribution of the data. Differences of qualitative data between the VDD group and CON group were analyzed using Chi-square test, whereas differences across the quartiles of TyG index were analyzed using Chi-square test for linear trend. The correlations between the TyG index and vitamin D or glycolipid-metabolic risk factors were determined using Spearman’s correlation analysis. The optimal cutoff points for the TyG index were determined using receiver operating characteristic (ROC) curve analysis. The relationships between the TyG index and VDD prevalence were analyzed by using four models of multivariate logistic regression analyses: (1) crude; (2) model 1: adjustments for age, sex, smoking, obesity, and hypertension; (3) model 2: further adjustments for other clinical parameters, including BMI, SBP, DBP, insulin therapy, hypoglycemic agents’ medication, and statin therapy; (4) model 3: further adjustments for laboratory variables, including HbA1c, TC, LDL-C, HDL-C, Scr, UA, and albumin. Further subgroup analyses were performed after stratifying the participants by age (≥65 or <65 years), sex (male or female), smoking (yes or no), and hypertension (yes or no). All variables were analyzed using SPSS (version 25.0, Inc, Chicago, IL, USA) and Graph Pad Prism software (version 7.0, Inc., LaJolla, CA, USA). A statistically significant difference was considered as two-tailed *p* < 0.05.

## 3. Results

### 3.1. Baseline Characteristics of the Participants

A total of 1034 participants with T2DM were enrolled, including 632 VDD patients and 402 controls. In the whole cohort, 621 were men (60.1%) and 413 were women (39.9%) and the average age was 62.0 ± 11.14 years. The interquartile range (25–75%) of serum 25(OH)D levels was 34.00–59.00 nmol/L (median 44.00 nmol/L) and the mean TyG index was 9.11 ± 0.75. The participants’ demographic and laboratory variables are shown in Table 1 according to the serum levels of 25(OH)D. Participants with a lower serum level of 25(OH)D were more obese and were more prone to have a history of hypertension, along with the therapy of insulin (all *p* < 0.05). They exhibited significantly higher levels of SBP, UACR, and TyG index, together with elevated serum levels of fasting glucose, HbA1c, TC, TG, and LDL-C, but lower Albumin levels (all *p* < 0.01). Unexpectedly, the VDD group had a reduced fasting glucose level (*p* = 0.05) and a lower percentage of the hypoglycemic agent usage (*p* = 0.01). No significant differences were observed in terms of age, sex, smoking status, statin use, DBP, C-peptide, and other investigated laboratory characteristics between the two groups (Table 1 and Figure 2a).

### 3.2. Correlation between the TyG Index and Vitamin D or Glycolipid-Metabolic Risk Factors

As shown Table 1 and Figure 2a, the VDD group showed significantly higher levels of TyG index values than the CON group (9.21 ± 0.77 vs. 8.96 ± 0.69, *p* < 0.05). Besides, the TyG index correlated negatively with serum 25(OH)D levels (Spearman’s rho = −0.172, *p* < 0.001, Figure 2b). Furthermore, TyG index was correlated with HbA1c (r = 0.300), C-peptide (r = 0.201), TC (r = 0.353), LDL-C (r = 0.348), HDL-C (r = −0.368), BMI (r = 0.241) (all *p* < 0.001), and DBP (r = 0.162, *p* = 0.001), while no significant correlation with SBP was observed (r = 0.028, *p* = 0.370) (Table 2).

### 3.3. Association between the TyG Index and VDD

All the participants were categorized into four groups according to the quartiles of the TyG index (Q1 (first quartile): TyG ≤ 8.57, Q2 (second quartile): 8.58 < TyG ≤ 9.03, Q3 (third quartile): 9.04 < TyG ≤ 9.53, Q4 (fourth quartile): TyG ≥ 9.54). The percentages of VDD prevalence were 48.8%, 58.6%, 64.7%, and 71.4% in the Q1, Q2, Q3, and Q4 quartiles, respectively (*p* for trend < 0.001, Figure 3). To determine a cutoff point of TyG index in relation to VDD, ROC analysis was used. It was found that TyG index is predictive for diagnosing the patients with VDD (AUC: 0.647, Youden’s index: 0.331, *p* < 0.001, Figure 4) with a high sensitivity (75.0%) and specificity (41.9%). TyG > 9.03 was determined as the optimal cutoff point for distinguishing 25(OH)D <50 nmol/L (Figure 4).

### 3.4. The Predictive Value of TyG Index for VDD Prevalence

To further explore whether the TyG index could be a predictor for VDD prevalence, multivariate logistic regression analyses were conducted. Taking the Q1 quartile of TyG index as a reference, an increasing trend of VDD prevalence was presented in the other three groups divided by TyG quartiles, which the odds ratio (OR) (95% confidence interval (CI)) were 1.561 (1.087–2.242), 2.011 (1.394–2.902) and 2.709 (1.849–3.969) in unadjusted analyses, respectively (all *p* < 0.05, Table 3). The association was still statistically significant after adjustments for age, sex, smoking, obesity, and hypertension (model 1). In addition, a slightly higher OR was shown after adjustments for other physical measurements and medication status, such as BMI, SBP, DBP, insulin therapy, hypoglycemic agents’ medication and statin therapy (model 2). Further adjustments for laboratory parameters, including HbA1c, TC, LDL-C, HDL-C, Scr, UA, and Alb plus model 2 covariates (model 3), produced similar results (OR (95% CI) for Q2: 1.708 (1.132–2.576), Q3: 2.041 (1.315–3.169), Q4: 2.543 (1.520–4.253), all *p* < 0.05, Table 3).

Stratifying the participants based on age (≥65 or <65 years), sex (male or female), smoking status (yes or no), and hypertension (yes or no), subgroup analyses remained in general consistent with the main findings (Figure 5). In addition, participants aged ≥65 years had higher risks than those aged <65 years for the prevalence of VDD, without statistically significant interactions (*p* for interaction = 0.999). For participants aged ≥65 years, the highest quartile of TyG index showed a 3.461-fold higher risk of VDD prevalence compared with the lowest quartile (OR: 3.461, 95% CI: 1.471–8.146, *p* = 0.004). However, no statistical significance was observed among the quartiles of TyG index in participants aged <65 years. Moreover, an increasing trend of VDD risk was observed in males in all quartiles of TyG index (OR (95% CI) for Q2: 1.786 (1.049–3.043), Q3: 2.191 (1.238–3.876), Q4: 2.596 (1.279–5.266), all *p* < 0.05, *p* for interaction = 0.447), while the risk was limited to the highest quartile of TyG index in females (OR (95% CI) for Q4: 2.263 (1.007–5.085), *p* = 0.048). Interestingly, when the participants were stratified by smoking status (*p* for interaction = 0.757), no significant association was shown between the TyG and the risk of VDD prevalence in smokers. However, an approximately two-fold higher risk of VDD prevalence was observed in all quartiles of TyG index in non-smokers (OR (95% CI) for Q2: 2.180 (1.315–3.615), Q3: 2.086 (1.222–3.563), Q4: 2.616 (1.410–4.854), all *p* < 0.01). When the participants were stratified by hypertension (*p* for interaction = 0.619), TyG index in the higher quartiles presented a higher risk of VDD prevalence in the patients with hypertension (OR (95% CI) for Q3: 3.144 (1.757–5.626), Q4: 2.780 (1.410–5.478), all *p* < 0.01). Conversely, the risk was only existed in the lower TyG index quartile in patients without hypertension (OR (95% CI) for Q2: 2.249 (1.113–4.546), *p* = 0.024) (Figure 5).

## 4. Discussion

In this study, significantly higher TyG index values were observed in patients with VDD than those with non-VDD in T2DM population. Besides, the TyG index correlated negatively to the 25(OH)D levels. Furthermore, we identified that the TyG index was a predictor for the diagnosis of VDD, manifested by an increased risk of VDD prevalence in line with the increase in TyG index quartiles. These results suggested that high TyG index levels may contribute to VDD pathogenesis in T2DM.

For most people, vitamin D is mainly obtained by cutaneous production from sunlight due to limited sources from natural diet. However, the amount and effectiveness of sunlight that reaches the skin is influenced by various factors, including latitude, altitude, season, race, clothing, sunscreen use, and age [38]. Vitamin D status is assessed by measurement of serum 25(OH)D level, and a serum 25(OH)D level below 50 nmol/L (20 ng/mL) is usually considered to be VDD [6,39]. VDD affects the entire world, as it was estimated that 60–80% of the population suffers from VDD or vitamin D insufficiency worldwide [40]. A large-scale cross-sectional study reported that about 75.2% of the adults suffered from VDD in Lanzhou (latitude 36° N), a city in northwestern China [34]. The prevalence of VDD was relatively lower in some southern cities of China, but still reached over 30% in Shanghai (latitude 31° N) and Guangzhou (latitude 23.1° N) [33,41]. A large number of studies demonstrated that VDD increased the risk of T2DM incidence [15,16,17,42,43,44]. A large-scale Danish population study reported a 1.12-fold increase in T2DM incidence with a 50% lower level of 25(OH)D [15]. Subsequent prospective cohort studies revealed the similar results [16,17,42,44]. In the present cross-sectional study, a prevalence of 61.1% was observed in the population from Changsha (latitude 28° N), a central city of China. the prevalence was a slightly lower than another study from the same city, which may be partly explained by different age, diet, and lifestyle [45].

The underlying reasons responsible for the association between VDD and T2DM have not been fully clarified. One of the key mechanisms may be that vitamin D could improve the glucose utilization in some tissues that are related to insulin responsiveness, such as adipose tissue and skeletal muscle [11,12]. It could alleviate IR in diabetes related obesity through various pathways like peroxisome proliferator-activated receptor delta that was co-expressed with VDR [46]. However, it remains controversial whether vitamin D has a great influence on the development of IR and T2DM. On the one hand, some studies from observational and longitudinal cohorts reported that serum 25(OH)D level was inversely related to IR and T2DM and increased the risk of T2DM incidence [13,14,15,16,17]. On the other hand, some randomized controlled trials of supplementing vitamin D on glucose metabolism yielded inconsistent results [47,48]. Gulseth et al. reported that vitamin D supplementation had no significant effect on insulin secretion and action in participants with moderated or no vitamin D deficiency. However, the study had very few subjects with severe vitamin D deficiency (defined as a baseline level ≤25 nmol/L), hence the potential effect of vitamin D supplementation on these subjects alone was not very clear [47]. However, Pittas et al. demonstrated a reduced risk in incident diabetes with vitamin D supplementation [48]. A review reported that, although the evidence from a series of randomized controlled trials did not support short-term vitamin D supplementation in the T2DM population, a beneficial effect of vitamin D supplementation was seen in patients with poorly controlled T2DM [49]. In the present observational study, we found a negative correlation between the TyG index and VVD, suggesting that IR is involved in the pathogenesis of VDD, which was consistent with some other observational studies. In the future, larger-scale, multi-center, randomized controlled trials are needed and the further mechanisms need to be clarified.

IR is a hallmark as well as a main pathophysiological mechanism of metabolic syndrome and T2DM [50]. However, methods to measure IR, such as euglycemic-hyperinsulinemic clamp test, are relatively invasive, complicated, and expensive [51]. Therefore, some non-invasive and inexpensive laboratory tests have been proposed to identify IR. The most commonly used method for evaluating insulin resistance is the homeostatic model assessment of insulin resistance (HOMA-IR), which is derived from the insulin and glucose in the fasting state [52]. However, insulin levels are not routinely tested in some medical institutions, especially in most small hospitals. There are some surrogate markers for IR, such as waist/hip ratio, C-peptide, visceral adiposity index, lipid accumulation product, and some lipid indicators. TyG index is a simpler and inexpensive biomarker for identifying IR where the measurement of insulin is not available [18]. Emerging evidence have revealed that TyG index level was associated positively with the risk of T2DM, diabetic complications, and metabolic syndrome [20,21,22,23,25,53]. Considering that TyG index and VDD were all closely related to IR in T2DM, it was hypothesized that TyG index may have an association with vitamin D levels and could be a predictor for VDD diagnosis in patients with T2DM.

As expected, we observed that the TyG index was correlated negatively to vitamin D levels, while correlated positively to glycolipid-metabolic risk factors, e.g., HbA1c, C-peptide, TC, LDL-C, SBP, and BMI. The results were consistent with the previous studies [26,54]. Jia et al. [26] reported a correlation coefficient of −0.13 between the TyG index and 25(OH)D, while the correlation coefficient was −0.172 in our study. Besides, when all the participants were categorized into four groups based on the TyG index quartiles, an increased trend of VDD prevalence was presented with the TyG index increase. Furthermore, we found the optimal cutoff point of the TyG index for the diagnosis of VDD was 9.03, which showed the highest sensitivity and specificity. These results suggested that TyG index could be a credible indicator for identifying VDD in patients with T2DM. Multivariate logistic regression analyses confirmed the TyG index’s predictive ability for VDD prevalence. Jia et al. [26] reported that serum vitamin D levels of male patients with T2DM were negatively associated with TyG index but not those of female patients, which was a little different from that in our study. We observed an increasing trend of VDD risk in all TyG index quartiles in males, while the risk only existed in the highest TyG index quartile in females. The causes of sex-differences may be associated with estrogens, as evidenced by a negative association between estrogen and serum TG levels [55,56]. This may partly explain why there was no obvious association between TyG index and VDD in female patients, but only in the highest TyG index quartile.

Intriguingly, subgroup analyses revealed a strong and significant association between the TyG index and VDD in participants aged over 65 years, but no significant difference was observed in the participants aged below 65 years. For participants aged ≥65 years, the highest quartile of TyG index showed a 3.461-fold higher risk of VDD prevalence compared with the lowest quartile. These results indicated that TyG index could be a more efficient predictor of VDD among elderly T2DM patients. Dhas et al. [57] reported a negative association between the TyG index and 25(OH)D levels in Indian middle-aged T2DM and healthy participants. A large-scale observational study using the data from the Comprehensive National Nutrition Survey found that the TyG index associated negatively with 25(OH) levels in Indian adolescent with or without T2DM [58]. However, the association of the TyG index and 25(OH) levels has not been studied in elderly T2DM patients. To our knowledge, this study firstly reported that TyG index could be a predictor for VDD prevalence in elderly T2DM patients.

There exist several limitations in this study. Firstly, it was an observational study in a central city of China. Thus, the conclusions are not comprehensive and need to be confirmed in subsequent large-scale and prospective studies. Secondly, the confounding factors, such as lipid lowering drugs, diet, season, clothing, sunscreen use, and physical activities, are not eliminated. Thirdly, the participants in this study were from a central city of China. Hence, the conclusions may not be applicable to other populations. Finally, the underlying mechanisms between the TyG index and vitamin D levels should be explored in the future.

## 5. Conclusions

Patients with higher TyG index were more likely to have an increased risk of VDD in the T2DM population. These findings suggested that IR is involved in the pathogenesis of VDD. Improvement of IR may have some implications for the prevention and treatment of VDD in the T2DM population.

## Figures and Tables

**Figure 1 nutrients-15-00639-f001:**
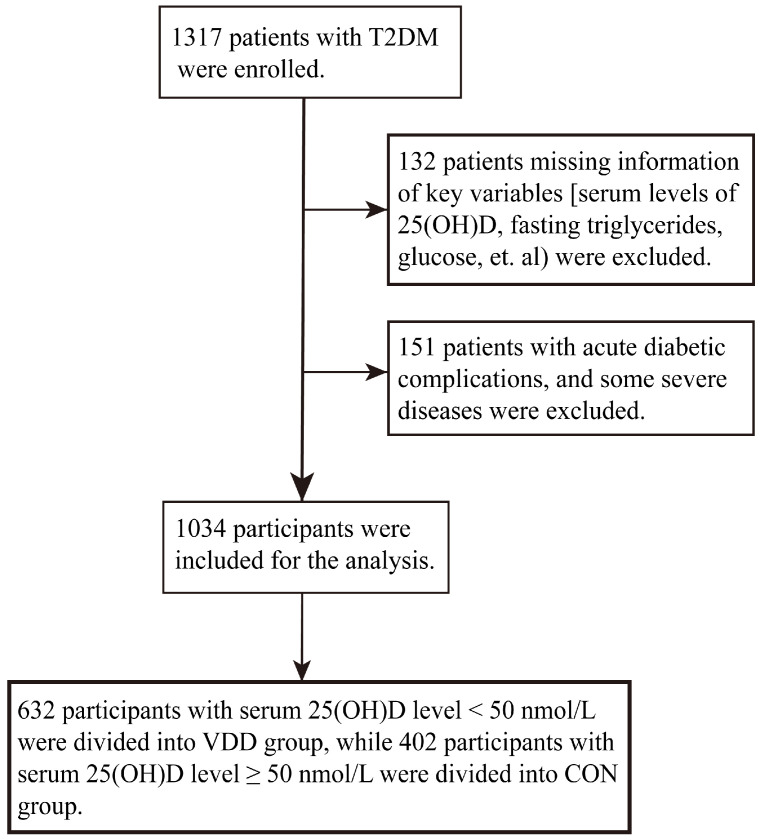
Flowchart of the participants included in the present study.

**Figure 2 nutrients-15-00639-f002:**
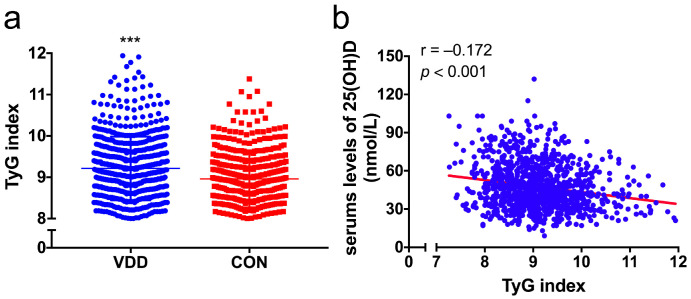
Comparison of the TyG index levels between the two groups (**a**) and the correlation between the TyG index and 25(OH)D (**b**). (**a**) comparisons of the TyG index were analyzed by Student’s *t*-test. *** *p* < 0.001 vs. the CON group. (**b**) The correlation between the levels of TyG index and 25(OH)D were analyzed using Spearman’s correlation analysis.

**Figure 3 nutrients-15-00639-f003:**
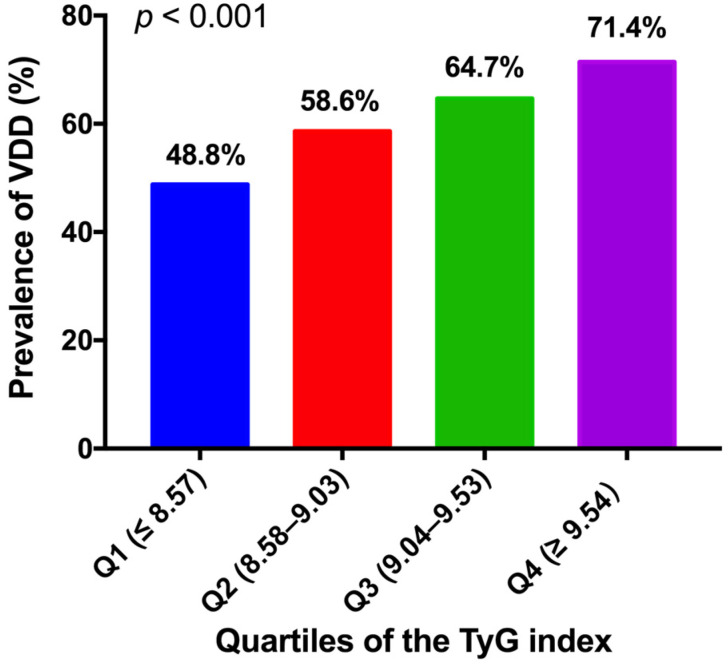
Prevalence of VDD across the TyG index quartiles. Comparisons of the percentages of VDD prevalence among the TyG index quartiles was performed using Chi-square test for linear trend test.

**Figure 4 nutrients-15-00639-f004:**
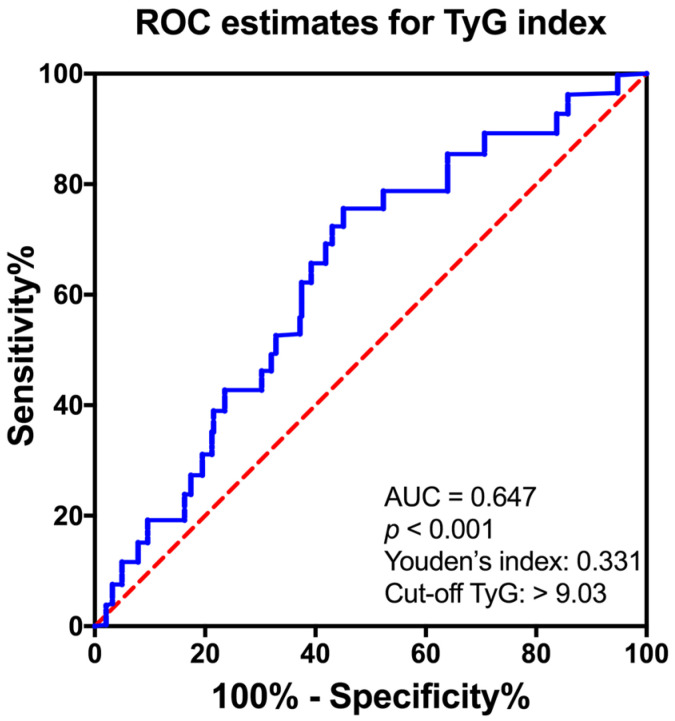
The optimal cutoff points of the TyG index for VDD diagnosis. Data were analyzed by ROC analysis. AUC values and Youden’s index are showed in brackets. Dotted line showed a random distribution.

**Figure 5 nutrients-15-00639-f005:**
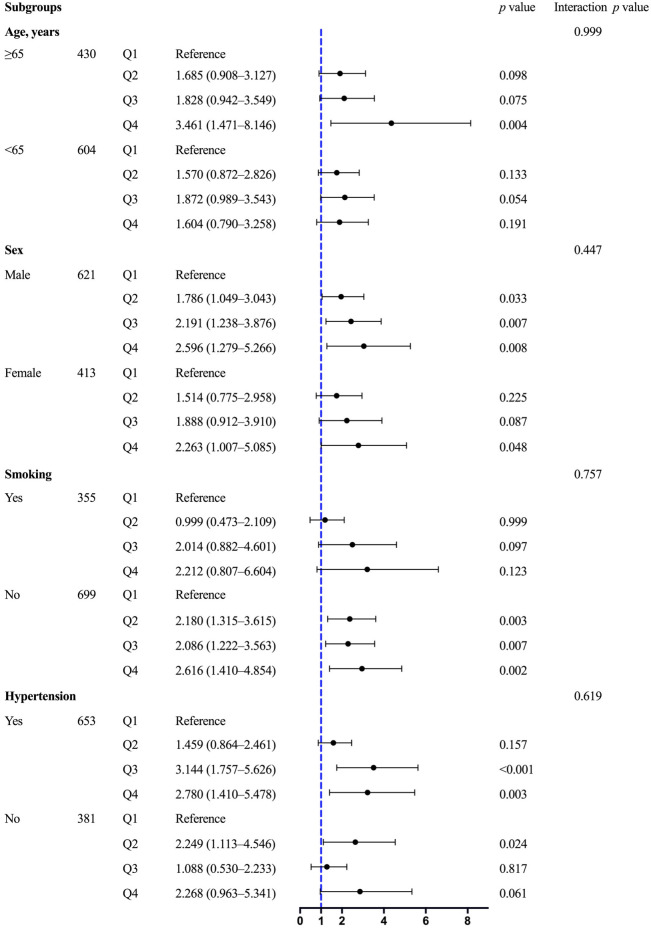
The association between the TyG index and VDD prevalence in subgroups. multivariate logistic regression analyses (adjustments for age, sex, smoking, obesity, hypertension, BMI, SBP, DBP, insulin therapy, hypoglycemic agents’ medication, statin therapy, HbA1c, TC, LDL-C, HDL-C, Scr, UA, and albumin) was used in subgroups based on age (≥65 or <65 years), sex (male or female), smoking (yes or no), hypertension (yes or no).

**Table 1 nutrients-15-00639-t001:** Baseline characteristics of the participants based on vitamin D levels.

	All (*n* = 1034)	VDD (*n* = 632)	CON (*n* = 402)	*p* Value
Age, years	62.0 ± 11.14	61.9 ± 11.38	62.2 ± 10.77	0.66
Male, n (%)	621 (60.1)	376 (59.5)	256 (40.5)	0.64
Smoking, n (%)	335 (32.4)	217 (34.4)	118 (29.4)	0.08
Obesity, n (%)	134 (13.0)	96 (15.2)	39 (9.7)	0.01
Hypertension, n (%)	653 (63.2)	415 (65.7)	238 (59.2)	0.04
Insulin therapy, n (%)	568 (54.9)	382 (60.4)	186 (46.3)	<0.001
Hypoglycemic agents, n (%)	916 (88.6)	546 (86.4)	368 (91.5)	0.01
Statin therapy, n (%)	757 (73.2)	469 (74.2)	288 (71.6)	0.36
SBP, mmHg	135.79 ± 20.40	137.16 ± 20.73	133.64 ± 19.71	0.007
DBP, mmHg	80.76 ± 11.83	80.86 ± 12.06	80.60 ± 11.47	0.73
BMI, kg/m^2^	24.28 ± 3.35	24.46 ± 3.38	24.00 ± 3.30	0.03
Fasting glucose, mmol/L	7.02 (5.80–8.64)	6.96 (5.76–8.75)	7.00 (5.90–8.84)	0.05
HbA1c, %	8.38 (6.93–10.20)	8.72 (7.11–10.66)	8.12 (6.89–9.70)	0.005
C-peptide, pmol/L	372.00 (224.00–574.93)	337.50 (208.20–572.00)	358.40 (223.00–549.00)	0.595
TC, mmol/L	4.40 ± 1.21	4.53 ± 1.29	4.19 ± 1.03	<0.001
TG, mmol/L	1.50 (1.02–2.23)	1.67 (1.06–2.38)	1.32 (0.91–1.92)	<0.001
LDL-C, mmol/L	2.78 ± 1.05	2.89 ± 1.10	2.60 ± 0.93	<0.001
HDL-C, mmol/L	1.08 ± 0.31	1.07 ± 0.31	1.09 ± 0.30	0.34
Albumin, g/L	37.81 ± 4.52	37.39 ± 4.76	38.47 ± 4.05	<0.001
UA, μmol/L	333.29 ± 108.05	337.20 ± 117.28	327.16 ± 91.54	0.14
BUN, mmol/L	6.48 (5.20–8.00)	6.67 (5.20–8.10)	6.40 (5.28–7.68)	0.25
Scr, μmol/L	75.00 (60.40–98.00)	76.00 (61.00–105.00)	76.00 (63.00–96.10)	0.92
eGFR, ml/min per 1.73 m^2^	84.04 (59.90–102.18)	82.82 (56.03–103.13)	85.46 (65.40–100.26)	0.40
UACR, mg/g	18.00 (5.25–147.00)	35.10 (7.00–309.60)	12.10 (4.30–82.50)	<0.001
25(OH)D, nmol/L	44.00 (34.00–59.00)	37.00 (31.00–43.00)	63.00 (56.00–71.00)	<0.001
TyG index	9.11 ± 0.75	9.21 ± 0.77	8.96 ± 0.69	<0.001

Values were expressed as mean ± standard deviation (SD) for normally distributed data or median (interquartile range) when abnormally distributed data, or *n* (%). Comparisons of the quantitative variables between the VDD group and CON group were analyzed by Student’s *t*-test or non-parametric Mann-Whitney U test according to the distribution of the data. Differences of qualitative data between the VDD group and CON group were analyzed using Chi-square test.

**Table 2 nutrients-15-00639-t002:** Spearman’s correlation between TyG index and glycolipid-metabolic risk factors.

	r	*p* Value
HbA1c, %	0.300	<0.001
C-peptide, pmol/L	0.201	<0.001
TC, mmol/L	0.353	<0.001
LDL-C, mmol/L	0.348	<0.001
HDL-C, mmol/L	−0.368	<0.001
BMI, kg/m^2^	0.241	<0.001
SBP, mmHg	0.028	0.370
DBP, mmHg	0.162	0.001

**Table 3 nutrients-15-00639-t003:** Logistic regression analyses showing the predictive value of TyG index for VDD prevalence.

	Unadjusted		Model 1		Model 2		Model 3	
	OR (95% CI)	*p* Value	OR (95% CI)	*p* Value	OR (95% CI)	*p* Value	OR (95% CI)	*p* Value
Q1	Reference		Reference		Reference		Reference	
Q2	1.486 (1.044–2.114)	0.028	1.561 (1.087–2.242)	0.016	1.747 (1.204–2.535)	0.003	1.708 (1.132–2.576)	0.011
Q3	1.921 (1.348–2.739)	<0.001	2.011 (1.394–2.902)	<0.001	2.162 (1.480–3.158)	<0.001	2.041 (1.315–3.169)	0.001
Q4	2.625 (1.822–3.782)	<0.001	2.709 (1.849–3.969)	<0.001	2.894 (1.940–4.319)	<0.001	2.543 (1.520–4.253)	<0.001

Model 1: adjustments for age, sex, smoking, obesity, and hypertension. Model 2: adjustments for model 1 covariates plus BMI, SBP, DBP, insulin therapy, hypoglycemic agents’ medication, and statin therapy. Model 3: adjustments for model 2 covariates plus HbA1c, TC, LDL-C, HDL-C, Scr, UA, and albumin.

## Data Availability

The data generated in the current study are available from the corresponding author on reasonable request. However, the availability of these data is limited and could be used under the permission of the current stud, and thus are not available to public.

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
