# Peer review of "Association between the Triglyceride-Glucose Index and Vitamin D Status in Type 2 Diabetes Mellitus"

_nutrients, 2023, doi:10.3390/nu15030639_

Round 1

Reviewer 1 Report

In this study, Xiang and co-workers explore whether an association exists between the TyG index and VDD in patients with T2DM and whether the TyG index has a predictive potential  for VDD prevalence. They demonstrate  that patients with VDD display a significantly higher TyG index than those with non-VDD in T2DM population and furthermore, the TyG index correlates negatively  with the 25(OH)D levels. Additionally, the authors identify the TyG index  as a predictor for the diagnosis of VDD. This study is important and  very interesting in the context of the significance of the TyG index in VDD. However, there are a few minor comments:

1. the flow chart presenting the procedure for assigning patients to the appropriate groups should be included;

2. in the discussion, the ethnicity of the population should be addressed as a limitation.  It make generalizing to other populations difficult;

3. please describe the statistical method used to calculate statistical significance below Table 1;

4. line 44, please add the abbreviation VDR after the full name of the receptor. Please replace "have been detected" with  "has been detected";

5. line 75, please provide protocol code and date of approval;

6. line 134, the word "rerolled" should be changed to "enrolled".

Author Response

Comments and Suggestions for Authors

Reviewer 1

In this study, Xiang and co-workers explore whether an association exists between the TyG index and VDD in patients with T2DM and whether the TyG index has a predictive potential for VDD prevalence. They demonstrate that patients with VDD display a significantly higher TyG index than those with non-VDD in T2DM population and furthermore, the TyG index correlates negatively with the 25(OH)D levels. Additionally, the authors identify the TyG index as a predictor for the diagnosis of VDD. This study is important and very interesting in the context of the significance of the TyG index in VDD. However, there are a few minor comments:

1. the flow chart presenting the procedure for assigning patients to the appropriate groups should be included;

Response: Thank you for your valuable advice, the flow chart has been added in Figure 1 and the related description was added in the revised manuscript (see figure 1 and line 78-83).

2. in the discussion, the ethnicity of the population should be addressed as a limitation.  It make generalizing to other populations difficult;

Response: Thank you for your careful reading and offering this kind suggestion. the ethnicity of the population has been added as a limitation in the discussion section of the revised manuscript (see line 351-352).

3. please describe the statistical method used to calculate statistical significance below Table 1;

Response: Thank you for your valuable suggestions. The description of the statistical method used in Table 1 has been added below Table 1 in the revised manuscript (see line 173-175).

4. line 44, please add the abbreviation VDR after the full name of the receptor. Please replace "have been detected" with "has been detected";

Response: Thank you for kind suggestion, and we are so sorry for not checking it carefully. The abbreviation VDR has been added, and “have been detected" has been replaced by “has been detected” in the revised manuscript (see line 44).

5. line 75, please provide protocol code and date of approval;

Response: Thank you for the sincere comments. The protocol code and date of approval have been added in the revised manuscript (see line 75).

6. line 134, the word "rerolled" should be changed to "enrolled".

Response: Thank you for your advice, and we are sorry for the spelling mistake. The word “enrolled” has been corrected in the revised manuscript (see line 157).

Reviewer 2 Report

The study of 1034  T2DM patients and the association between TyG index and VVD is interesting. In the study one observe a negative correlation between TyG and VVD and conclude that IR is associated with VDD in patients with T2DM. 

The hypothesis that Vitamin D has great influence on the development of IR and T2DM is highly debatable ( see: DOI: 10.2337/dc16-2302 and DOI: 10.1210/clinem/dgaa594).  Also on may speculate that the observation found is merely a surrogate for higher IR in a part of a population of T2DM with poorer nutrition and hence lower Vitamin D. This last point should have been addressed in the study.

Major concern:

The researchers use < 50nmol/l for VDD, but what is the actual reference values (normal values) in the actual population?

Is T2DM an immune inflammatory disease as postulated in the introduction?

Has TyG index been evaluated in the actual population as a marker for IR?

Is these findings actually suggestive of an influence of IR on VDD, is not this vice versa? Lower Vitamin D due to poor nutrition in the population with higher IR?

Minor concerns:

Is it not better to just measure Vitamin D than to use TyG index cut off  to find VDD?

Why is the population suddenly divided in to case and control in the result part?

In table 1, why is the upper limit in the VDD group not 50 nmol/l for vitamin D?

Is really a high TyG index indicative of the VDD pathogenesis in T2DM? And dose such an pathogenesis exist?

Why is not waist / hip ratio or fasting C-peptide mentioned as surrogates for IR?

Author Response

Comments and Suggestions for Authors

The study of 1034 T2DM patients and the association between TyG index and VVD is interesting. In the study one observe a negative correlation between TyG and VVD and conclude that IR is associated with VDD in patients with T2DM. The hypothesis that Vitamin D has great influence on the development of IR and T2DM is highly debatable (see: DOI: 10.2337/dc16-2302 and DOI: 10.1210/clinem/dgaa594).  Also on may speculate that the observation found is merely a surrogate for higher IR in a part of a population of T2DM with poorer nutrition and hence lower Vitamin D. This last point should have been addressed in the study.

Response: Thank you for your careful reading of our manuscript and we are extremely grateful to you for your valuable and insightful comments. We agree with you that it is controversial that vitamin D has a great influence on the development of IR and T2DM. On the one hand, some studies from observational and longitudinal cohort reported that serum 25(OH)D level was inversely related to IR and T2DM, and increased the risk of T2DM incidence[1-5]. On the other hand, some randomized controlled trials of supplementing vitamin D on glucose metabolism yielded inconsistent results[6, 7]. Gulseth et al. reported that vitamin D supplementation had no significant effect on insulin secretion and action in participants with moderated or no VDD. But the study had very few subjects with severe VDD (defined as a baseline level ≤25 nmol/L), hence the potential effect of vitamin D supplementation on these subjects alone was not very clear[6]. However, Pittas et al. demonstrated a reduced risk in incident diabetes with vitamin D supplementation[7]. A review reported that although the evidence from a series of randomized controlled trials did not support short-term vitamin D supplementation in T2DM population, a beneficial effect of vitamin D supplementation is seen in patients with poorly controlled T2DM[8]. In the present observational study, we found a negative correlation between the TyG index and VVD, suggesting that IR is involved in the pathogenesis of VDD, which was consistent with some other observational studies. In the future, more large-scale multi-center randomized controlled trials are needed and the further mechanisms need to be clarified. According to your suggestive advice, we have added this part in the discussion section in the revised manuscript (line 280-298).

Major concern:

1. The researchers use < 50nmol/l for VDD, but what is the actual reference values (normal values) in the actual population?

Response: Thank you for the sincere comments about this issue, and we are sorry for not elucidating it clearly. In a large-scale cross-sectional study, a serum level of 25(OH)D below 20 ng/mL (50 nmol/L) was considered as VDD, and the prevalence of VDD among urban Beijing, the capital of China, was 87.1%[9]. Besides, Lu et al. demonstrated that the prevalence of VDD [defined as serum 25(OH)D level < 20 ng/mL] was 30% in males and 46% in females in the population of Shanghai, a central city of China[10]. Another large-scale cross-sectional study reported that about 75.2% of the adults suffered from VDD [defined as serum 25(OH)D level < 20 ng/mL] in Lanzhou, a city in northwestern China[11]. Thank you again for your constructive comments. As you suggested, the references have been added in the revised manuscript (see references 32-34).

2. Is T2DM an immune inflammatory disease as postulated in the introduction?

Response: Thank you for the sincere comments about this issue. We searched some literatures and found that a chronic low-grade inflammation and an activation of the immune system were involved in the pathogenesis of T2DM[12, 13]. Postprandial hyperglycemia activates the inflammasome in peritoneal macrophages that secretes IL-1β. And then IL-1β contributed to the postprandial stimulation of insulin secretion, thereby establishing a feedforward loop[14]. These evidences demonstrated that innate immune system was involved in T2DM, but the description of immune disease may be not accurate for T2DM. Hence, in order to avoid misunderstanding, we have changed the sentence “Type 2 diabetes mellitus (T2DM) is an immune–inflammatory and age-related disease, ……” into “Type 2 diabetes mellitus (T2DM) is a low-grade inflammatory and age-related disease, ……” (see line 33-34).

3. Has TyG index been evaluated in the actual population as a marker for IR?

Response: Thank you for your valuable and thoughtful comments. Some non-invasive and inexpensive laboratory tests have been proposed to evaluate IR, with the HOMA-IR is the most commonly used. However, the insulin levels are not routinely tested in some medical institutions, especially in most essential hospitals. TyG index is a simpler and more inexpensive biomarker for identifying IR where the measurement of insulin is not available. In Chinese population, the application of the TyG index as a marker for IR has been evaluate in some studies. Du et al. reported that TyG index had the most significant association with HOMA-IR than visceral adiposity index (VAI), lipid accumulation product (LAP), and some lipid indicators, and is a better marker for early identification of IR individuals[15]. Besides, Huang et al. found a significant positive correlation between HOMA-IR and TyG index, and demonstrated that the TyG index is a simple, relatively accurate, clinically available surrogate marker of IR in middle-aged population of China[16]. In addition, Wang et al. reported that the TyG index is independently and more strongly associated with arterial stiffness in patients with T2DM when compared with the HOMA-IR[17]. Moreover, it has been revealed that the TyG index was associated positively with the risk of T2DM, diabetic complications in Chinese population[18-22]. Based on these evidences, the TyG index could be a surrogate marker of IR in Chines population. Thereby, we used TyG index as a marker for IR in the present study.

4. Is these findings actually suggestive of an influence of IR on VDD, is not this vice versa? Lower Vitamin D due to poor nutrition in the population with higher IR?

Response: Thank you for your thoughtful comments. In the present study, we observed a significantly higher level of the TyG index in VDD group. Further analysis revealed an increasing trend of VDD prevalence in line with the increase in TyG index quartiles. (Figure 3, Table 1, Table 3). These results suggested that TyG index, a surrogate marker for IR, may have an effect on VDD. Since it was an observational study, we could not draw a causal conclusion between IR and VDD, but only correlation between them. The results (Figure 3) also suggested a lower Vitamin D level in subjects with higher IR, but the underlying mechanisms remain not fully elucidated.

Minor concerns:

1. Is it not better to just measure Vitamin D than to use TyG index cut off  to find VDD?

Response: Thank you for your kind suggestion. Currently, the commonly used method to measure serum 25(OH)D level is chemiluminescence assay. However, it was not routinely detected in some medical institutions, especially in most essential hospitals in China. But the levels of blood lipids are routinely measured. So, we wondered if TyG index could be predictive for diagnosing the patients with VDD.

2. Why is the population suddenly divided in to case and control in the result part? Response: We are extremely grateful to you for pointing out this problem. It was known that both TyG index and VDD were closely related to IR in T2DM, thus we hypothesized that TyG index may have an association with vitamin D level. In the present study, we did observe a negative association between the TyG index and vitamin D level, and we further intended to evaluate if the TyG index could be used as a predictive tool for VDD prevalence. As expected, the results from multivariate logistic regression analyses confirmed an increased VDD prevalence in participants with higher TyG index. Thus, we divided the patients into VDD group and CON group for the analysis.

3. In table 1, why is the upper limit in the VDD group not 50 nmol/l for vitamin D?

Response: Thank you for your careful reading of our manuscript. In table 1, the parameters of abnormal distributed were presented as median with interquartile ranges (25–75%). Hence, “43.00 nmol/l” was not the upper limit in the VDD group but the third quartile. We are sorry for not clarifying it clearly, and have supplemented it in the revised manuscript (see line 135 and line 159).

4. Is really a high TyG index indicative of the VDD pathogenesis in T2DM? And dose such an pathogenesis exist?

Response: We are extremely grateful to you for pointing out this problem. The results in the present observational study found that high TyG index is associated with increased VDD prevalence. Based on these results, it was speculated that high TyG index representing IR may be involved in the pathogenesis of VDD in T2DM. However, the underlying mechanisms were not elucidated. Numerous studies have reported the association between VDD and IR, indicating that Vitamin D was involved in IR and T2DM. However, it was not clear whether IR has an effect on VDD. A review has reported that obese IR state was often associated with low circulating concentration of vitamin 25(OH)D[23]. The mechanisms for the VDD in the obese IR state may be related to impaired vitamin D metabolism[24], vitamin D sequestration[25], and impaired vitamin D release[26].

5. Why is not waist / hip ratio or fasting C-peptide mentioned as surrogates for IR?

Response: Thank you for your kind suggestion. As mentioned in response to the third question in major concerns, the TyG index is a surrogate marker of IR in Chines population. We agree with you that waist/hip ratio and fasting C-peptide are also surrogate markers for IR, but we mainly focused on the association between the TyG index on and vitamin D status in T2DM in the present study. Thank you again for this valuable suggestion, and we have supplemented it in the discussion section in the revised manuscript (see line 306-308).

References

  1. Scragg R, Sowers M, Bell C, Third National H and Nutrition Examination S. Serum 25-hydroxyvitamin D, diabetes, and ethnicity in the Third National Health and Nutrition Examination Survey. Diabetes Care. 2004; 27(12):2813-2818.
  2. Hurskainen AR, Virtanen JK, Tuomainen TP, Nurmi T and Voutilainen S. Association of serum 25-hydroxyvitamin D with type 2 diabetes and markers of insulin resistance in a general older population in Finland. Diabetes Metab Res Rev. 2012; 28(5):418-423.
  3. Afzal S, Bojesen SE and Nordestgaard BG. Low 25-hydroxyvitamin D and risk of type 2 diabetes: a prospective cohort study and metaanalysis. Clin Chem. 2013; 59(2):381-391.
  4. Gagnon C, Lu ZX, Magliano DJ, Dunstan DW, Shaw JE, Zimmet PZ, Sikaris K, Grantham N, Ebeling PR and Daly RM. Serum 25-hydroxyvitamin D, calcium intake, and risk of type 2 diabetes after 5 years: results from a national, population-based prospective study (the Australian Diabetes, Obesity and Lifestyle study). Diabetes Care. 2011; 34(5):1133-1138.
  5. Pittas AG, Sun Q, Manson JE, Dawson-Hughes B and Hu FB. Plasma 25-hydroxyvitamin D concentration and risk of incident type 2 diabetes in women. Diabetes Care. 2010; 33(9):2021-2023.
  6. Gulseth HL, Wium C, Angel K, Eriksen EF and Birkeland KI. Effects of Vitamin D Supplementation on Insulin Sensitivity and Insulin Secretion in Subjects With Type 2 Diabetes and Vitamin D Deficiency: A Randomized Controlled Trial. Diabetes Care. 2017; 40(7):872-878.
  7. Pittas AG, Jorde R, Kawahara T and Dawson-Hughes B. Vitamin D Supplementation for Prevention of Type 2 Diabetes Mellitus: To D or Not to D? J Clin Endocrinol Metab. 2020; 105(12):3721-3733.
  8. Krul-Poel YH, Ter Wee MM, Lips P and Simsek S. MANAGEMENT OF ENDOCRINE DISEASE: The effect of vitamin D supplementation on glycaemic control in patients with type 2 diabetes mellitus: a systematic review and meta-analysis. Eur J Endocrinol. 2017; 176(1):R1-R14.
  9. Ning Z, Song S, Miao L, Zhang P, Wang X, Liu J, Hu Y, Xu Y, Zhao T, Liang Y, Wang Q, Liu L, Zhang J, et al. High prevalence of vitamin D deficiency in urban health checkup population. Clin Nutr. 2016; 35(4):859-863.
  10. Lu HK, Zhang Z, Ke YH, He JW, Fu WZ, Zhang CQ and Zhang ZL. High prevalence of vitamin D insufficiency in China: relationship with the levels of parathyroid hormone and markers of bone turnover. PLoS One. 2012; 7(11):e47264.
  11. Zhen D, Liu L, Guan C, Zhao N and Tang X. High prevalence of vitamin D deficiency among middle-aged and elderly individuals in northwestern China: its relationship to osteoporosis and lifestyle factors. Bone. 2015; 71:1-6.
  12. Esser N, Legrand-Poels S, Piette J, Scheen AJ and Paquot N. Inflammation as a link between obesity, metabolic syndrome and type 2 diabetes. Diabetes Res Clin Pract. 2014; 105(2):141-150.
  13. Donath MY, Dinarello CA and Mandrup-Poulsen T. Targeting innate immune mediators in type 1 and type 2 diabetes. Nat Rev Immunol. 2019; 19(12):734-746.
  14. Dror E, Dalmas E, Meier DT, Wueest S, Thevenet J, Thienel C, Timper K, Nordmann TM, Traub S, Schulze F, Item F, Vallois D, Pattou F, et al. Postprandial macrophage-derived IL-1beta stimulates insulin, and both synergistically promote glucose disposal and inflammation. Nat Immunol. 2017; 18(3):283-292.
  15. Du T, Yuan G, Zhang M, Zhou X, Sun X and Yu X. Clinical usefulness of lipid ratios, visceral adiposity indicators, and the triglycerides and glucose index as risk markers of insulin resistance. Cardiovasc Diabetol. 2014; 13:146.
  16. Huang R, Cheng Z, Jin X, Yu X, Yu J, Guo Y, Zong L, Sheng J, Liu X and Wang S. Usefulness of four surrogate indexes of insulin resistance in middle-aged population in Hefei, China. Ann Med. 2022; 54(1):622-632.
  17. Wang S, Shi J, Peng Y, Fang Q, Mu Q, Gu W, Hong J, Zhang Y and Wang W. Stronger association of triglyceride glucose index than the HOMA-IR with arterial stiffness in patients with type 2 diabetes: a real-world single-centre study. Cardiovasc Diabetol. 2021; 20(1):82.
  18. Zhang M, Wang B, Liu Y, Sun X, Luo X, Wang C, Li L, Zhang L, Ren Y, Zhao Y, Zhou J, Han C, Zhao J, et al. Cumulative increased risk of incident type 2 diabetes mellitus with increasing triglyceride glucose index in normal-weight people: The Rural Chinese Cohort Study. Cardiovasc Diabetol. 2017; 16(1):30.
  19. Xuan X, Hamaguchi M, Cao Q, Okamura T, Hashimoto Y, Obora A, Kojima T, Fukui M, Yuan G, Guo Z, Luo Z, Qin Y, Luo X, et al. U-shaped association between the triglyceride-glucose index and the risk of incident diabetes in people with normal glycemic level: A population-base longitudinal cohort study. Clin Nutr. 2021; 40(4):1555-1561.
  20. Zou S, Yang C, Shen R, Wei X, Gong J, Pan Y, Lv Y and Xu Y. Association Between the Triglyceride-Glucose Index and the Incidence of Diabetes in People With Different Phenotypes of Obesity: A Retrospective Study. Front Endocrinol (Lausanne). 2021; 12:784616.
  21. Chen CL, Liu L, Lo K, Huang JY, Yu YL, Huang YQ and Feng YQ. Association Between Triglyceride Glucose Index and Risk of New-Onset Diabetes Among Chinese Adults: Findings From the China Health and Retirement Longitudinal Study. Front Cardiovasc Med. 2020; 7:610322.
  22. Pan Y, Zhong S, Zhou K, Tian Z, Chen F, Liu Z, Geng Z, Li S, Huang R, Wang H, Zou W and Hu J. Association between Diabetes Complications and the Triglyceride-Glucose Index in Hospitalized Patients with Type 2 Diabetes. J Diabetes Res. 2021; 2021:8757996.
  23. Pramono A, Jocken JWE and Blaak EE. Vitamin D deficiency in the aetiology of obesity-related insulin resistance. Diabetes Metab Res Rev. 2019; 35(5):e3146.
  24. Wamberg L, Christiansen T, Paulsen SK, Fisker S, Rask P, Rejnmark L, Richelsen B and Pedersen SB. Expression of vitamin D-metabolizing enzymes in human adipose tissue -- the effect of obesity and diet-induced weight loss. Int J Obes (Lond). 2013; 37(5):651-657.
  25. Wortsman J, Matsuoka LY, Chen TC, Lu Z and Holick MF. Decreased bioavailability of vitamin D in obesity. Am J Clin Nutr. 2000; 72(3):690-693.
  26. Di Nisio A, De Toni L, Sabovic I, Rocca MS, De Filippis V, Opocher G, Azzena B, Vettor R, Plebani M and Foresta C. Impaired Release of Vitamin D in Dysfunctional Adipose Tissue: New Cues on Vitamin D Supplementation in Obesity. J Clin Endocrinol Metab. 2017; 102(7):2564-2574.